# Learning Temporally Extended Skills in Continuous Domains as Symbolic Actions for Planning

**Jan Achterhold**       **Markus Krimmel**       **Joerg Stueckler**
Embodied Vision Group, Max Planck Institute for Intelligent Systems, Tübingen, Germany
{jan.achterhold,markus.krimmel,joerg.stueckler}@tuebingen.mpg.de

**Abstract:** Problems which require both long-horizon planning and continuous control capabilities pose significant challenges to existing reinforcement learning agents. In this paper we introduce a novel hierarchical reinforcement learning agent which links temporally extended skills for continuous control with a forward model in a symbolic discrete abstraction of the environment's state for planning. We term our agent SEADS for Symbolic Effect-Aware Diverse Skills. We formulate an objective and corresponding algorithm which leads to unsupervised learning of a diverse set of skills through intrinsic motivation given a known state abstraction. The skills are jointly learned with the symbolic forward model which captures the effect of skill execution in the state abstraction. After training, we can leverage the skills as symbolic actions using the forward model for long-horizon planning and subsequently execute the plan using the learned continuous-action control skills. The proposed algorithm learns skills and forward models that can be used to solve complex tasks which require both continuous control and long-horizon planning capabilities with high success rate. It compares favorably with other flat and hierarchical reinforcement learning baseline agents and is successfully demonstrated with a real robot. Project page: https://seads.is.tue.mpg.de

**Keywords:** temporally extended skill learning, hierarchical reinforcement learning, diverse skill learning

## 1   Introduction

Reinforcement learning (RL) agents have been applied to difficult continuous control and discrete planning problems such as the DeepMind Control Suite [1], StarCraft II [2], or Go [3] in recent years. Despite this tremendous success, tasks which require both continuous control capabilities and long-horizon discrete planning are classically approached with task and motion planning [4]. These problems still pose significant challenges to RL agents [5]. An exemplary class of environments which require both continuous-action control and long-horizon planning are *physically embedded games* as introduced by [5]. In these environments, a board game is embedded into a physical manipulation setting. A move in the board game can only be executed indirectly through controlling a physical manipulator such as a robotic arm. We simplify the setting of [5] and introduce physically embedded *single-player* board games which do not require to model the effect of an opponent. Our experiments support the findings of [5] that these environments are challenging to solve for existing flat and hierarchical RL agents. In this paper, we propose a novel hierarchical RL agent for such environments which learns skills and their effects in a known symbolic abstraction of the environment.

As a concrete example for a proposed embedded single-player board game we refer to the *LightsOutJaco* environment (see Fig. 1). Pushing a field on the *LightsOut* board toggles the illumination state (*on* or *off*) of the field and its non-diagonal neighboring fields. A field on the board can only be pushed by the end effector of the *Jaco* robotic arm. The goal is to reach a board state in which all fields are *off*. The above example also showcases the two concepts of *state* and *action* abstraction in decision making [6]. A state abstraction function $\Phi(s_t)$ only retains information in state $s_t$ which is relevant for a particular decision making task. In the LightsOut example, to decide which move to perform next (i.e., which field to push), only the illumination state of the board is relevant. A

6th Conference on Robot Learning (CoRL 2022), Auckland, New Zealand.

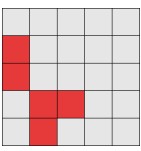 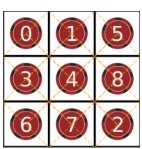 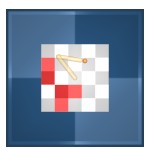 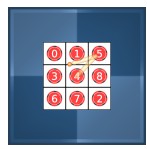 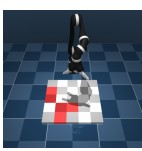 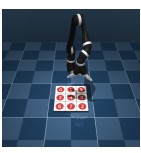

LightsOutCursor  TileSwapCursor  LightsOutReacher  TileSwapReacher  LightsOutJaco  TileSwapJaco

Figure 1: *LightsOut* (with on (red) and off (gray) fields) and *TileSwap* (fields in a rhombus are swapped if pushed inside) board games embedded into physical manipulation settings. A move in the board game can only indirectly be executed through controlling a manipulator.

*move* can be considered an *action abstraction*: A skill, i.e. high-level action (e.g., push top-left field), comprises a sequence of low-level actions required to control the robotic manipulator.

We introduce a two-layer hierarchical agent which assumes a discrete state abstraction $z_t = \Phi(s_t) \in \mathcal{Z}$ to be known and observable in the environment, which we in the following refer to as *symbolic observation*. In our approach, we assume that state abstractions can be defined manually for the environment. For LightsOut, the symbolic observation corresponds to the state of each field (on/off). We provide the state abstraction as prior knowledge about the environment and assume that skills induce changes of the abstract state. Our approach then learns a diverse set of skills for the given state abstraction as action abstractions and a corresponding forward model which predicts the effects of skills on abstract states. In board games, these abstract actions relate to *moves*. We jointly learn the predictive forward model $q_\theta$ and *skill policies* $\pi(a \mid s_t, k)$ for low-level control through an objective which maximizes the number of symbolic states reachable from any state of the environment (diversity) and the predictability of the effect of skill execution. Please see Fig. 2 for an illustration of the introduced temporal and symbolic hierarchy. The forward model $q_\theta$ can be leveraged to plan a sequence of skills to reach a particular state of the board (e.g., all fields off), i.e. to solve tasks. We evaluate our approach using two single-player board games in environments with varying complexity in continuous control. We demonstrate that our agent learns skill policies and forward models suitable for solving the associated tasks with high success rate and compares favorably with other flat and hierarchical RL baseline agents. We also demonstrate our agent playing LightsOut with a real robot.

In summary, we contribute the following: (1) We formulate a novel RL algorithm which, based on a state abstraction of the environment and an information-theoretic objective, jointly learns a diverse set of continuous-action skills and a forward model capturing the temporally abstracted effect of skill execution in symbolic states. (2) We demonstrate the superiority of our approach compared to other flat and hierarchical baseline agents in solving complex physically-embedded single-player games, requiring high-level planning and continuous control capabilities. We provide additional materials, including video and code, at `https://seads.is.tue.mpg.de`.

## 2   Related work

**Diverse skill learning and skill discovery.** Discovering general skills to control the environment through exploration without task-specific supervision is a fundamental challenge in RL research. DIAYN [7] formulates skill discovery using an information-theoretic objective as reward. The agent learns a skill-conditioned policy for which it receives reward if the target states can be well predicted from the skill. VALOR [8] proposes to condition the skill prediction model on the complete trajectory of visited states. Warde-Farley et al. [9] train a goal-conditioned policy to reach diverse states in the environment. Variational Intrinsic Control [10] proposes to use an information-theoretic objective to learn a set of skills which can be identified from their initial and target states. Relative Variational Intrinsic Control [11] seeks to learn skills relative to their start state, aiming to avoid skill representations that merely tile the state space into goal state regions. Both approaches do not learn a forward model on the effect of skill execution like our approach. Sharma et al. [12] propose a model-based RL approach (DADS) which learns a set of diverse skills and their dynamics models using mutual-information-based exploration. While DADS learns skill dynamics as immediate behavior $q(s_{t+1}|s_t, k)$, we learn a transition model on the effect of skills $q(z_T|z_0, k)$ in a symbolic abstraction, thereby featuring temporal abstraction.

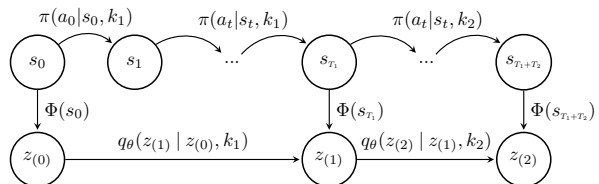

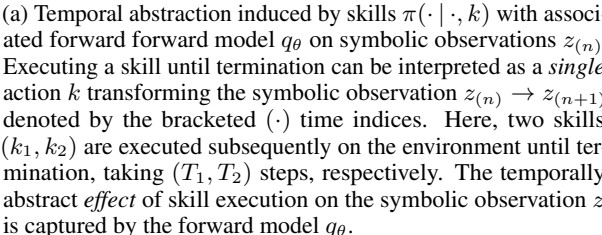

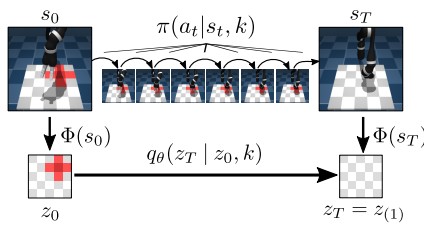

(a) Temporal abstraction induced by skills $\pi(\cdot \mid \cdot, k)$ with associated forward forward model $q_\theta$ on symbolic observations $z_{(n)}$. Executing a skill until termination can be interpreted as a *single* action $k$ transforming the symbolic observation $z_{(n)} \to z_{(n+1)}$ denoted by the bracketed $(\cdot)$ time indices. Here, two skills $(k_1, k_2)$ are executed subsequently on the environment until termination, taking $(T_1, T_2)$ steps, respectively. The temporally abstract *effect* of skill execution on the symbolic observation $z$ is captured by the forward model $q_\theta$.

(b) Symbolic abstraction $\Phi$ and temporal skill abstraction demonstrated on the `LightsOutJaco` environment. The symbolic observation $z$ represents the discrete state of the board while $s$ contains both the board state and the state of the *Jaco* manipulator. Executing skill $k$ by applying the skill policy $\pi(\cdot|\cdot, k)$ until termination leads to a change of the state of the board, which is modeled by $q_\theta$ with a single action $k$.

Figure 2: We aim to learn skills with associated policies $\pi(a_t|s_t, k)$ which lead to diverse and predictable (by the forward model $q_\theta$) transitions in a symbolic abstraction $z = \Phi(s)$ of the state $s$.

**Hierarchical RL.** Hierarchical RL can overcome sparse reward settings and time extended tasks by breaking the task down into subtasks. Some approaches such as methods based on MAXQ [13, 14] assume prior knowledge on the task-subtask decomposition. In SAC-X [15], auxiliary tasks assist the agent in learning sparse reward tasks and hierarchical learning involves choosing between tasks. Florensa et al. [16] propose to learn a span of skills using stochastic neural networks for representing policies. The policies are trained in a task-agnostic way using a measure of skill diversity based on mutual information. Specific tasks are then tackled by training an RL agent based on the discovered skills. Feudal approaches [17] such as HIRO [18] and HAC [19] train a high-level policy to provide subgoals for a low-level policy. In our method, we impose that a discrete state-action representation exists in which learned skills are discrete actions, and train the discrete forward model and the continuous skill policies jointly. Several approaches to hierarchical RL are based on the options framework [20] which learns policies for temporally extended actions in a two-layer hierarchy. Learning in the options framework is usually driven by task rewards. Recent works extend the framework to continuous spaces and discovery of options (e.g. [21, 22]). HiPPO [23] develop an approximate policy gradient method for hierarchies of actions. HIDIO [24] learns task-agnostic options using a measure of diversity of the skills. In our approach, we also learn task-agnostic (for the given state abstraction) hierarchical representations using a measure of intrinsic motivation. However, an important difference is that we do not learn high-level policies over options using task rewards, but learn a skill-conditional forward model suitable for planning to reach a symbolic goal state. Jointly, continuous policies are learned which implement the skills. Several approaches combine symbolic planning in a given domain description (state and action abstractions) with RL to execute the symbolic actions [25, 26, 27, 28, 29]. Similar to our approach, the work in Guan et al. [29] learns low-level skill policies using an information-theoretic diversity measure which implement known symbolic actions. Differently, we learn the action abstraction and low-level skills given the state abstraction.

**Representation learning for symbolic planning.** Some research has been devoted to learning representations for symbolic planning. Konidaris et al. [30] propose a method for acquiring a symbolic planning domain from a set of low-level options which implement abstract symbolic actions. In [31] the approach is extended to learning symbolic representations for families of SMDPs which describe options in a variety of tasks. Our approach learns action abstractions as a set of diverse skills given a known state abstraction and a termination condition which requires abstract actions to change abstract states. Toro Icarte et al. [32] learn structure and transition models of finite state machines through RL. Ugur and Piater [33] acquire symbolic forward models for a predefined low-level action repertoire in a robotic manipulation context. Chitnis et al. [34] concurrently learn transition models on the symbolic and low levels from demonstrations provided in the form of hand-designed policies, and use the learned models for bilevel task and motion planning. The approach also assumes the state abstraction function to be known. In [35] a different setting is considered in which the symbolic transition model is additionally assumed known and skill policies that execute symbolic actions

are learned from demonstrations. Other approaches such as DeepSym [36] or LatPlan [37] learn mappings of images to symbolic states and learn action-conditional forward models. In [37] symbolic state-action representations are learned from image observations of discrete abstract actions (e.g. moving puzzle tiles to discrete locations) which already encode the planning problem. Our approach concurrently learns a diverse set of skills (discrete actions) based on an information-theoretic intrinsic reward and the symbolic forward model. Differently, in our approach low-level actions are continuous.

## 3  Method

Our goal is to learn a hierarchical RL agent which (i) enables high-level, temporally abstract planning to reach a particular goal configuration of the environment (as given by a symbolic observation) and (ii) features continuous control policies to execute the high-level plan. Let $\mathcal{S}, \mathcal{A}$ denote the state and action space of an environment, respectively. In general, by $\mathcal{Z} = \{0, 1\}^D$ we denote the space of discrete symbolic environment observations $z \in \mathcal{Z}$ and assume the existence of a state abstraction $\Phi : \mathcal{S} \rightarrow \mathcal{Z}$. The dimensionality of the symbolic observation $D$ is environment-dependent. For the *LightsOutJaco* environment, the state $s = [q, \dot{q}, z] \in \mathcal{S}$ contains the robot arms' joint positions and velocities $(q, \dot{q})$ and a binary representation of the board $z \in \{0, 1\}^{5 \times 5}$. The action space $\mathcal{A}$ is equivalent to the action space of the robotic manipulator. In the *LightsOutJaco* example, it contains the target velocity of all actuable joints. The discrete variable $k \in \mathcal{K} = \{1, ..., K\}$ refers to a particular *skill*, which we will detail in the following. The number of skills $K$ needs to be set in advance, but can be chosen larger than the number of actual skills.

We equip our agent with symbolic planning and plan execution capabilities through two components: First, a forward model $\hat{z} = f(z, k) = \text{argmax}_{z'} q_\theta(z' \mid z, k)$ allows to *enumerate* all possible symbolic successor states $\hat{z}$ of the current symbolic state $z$ by iterating over the discrete variable $k$. This allows for node expansion in symbolic planners. Second, a family of discretely indexed policies $\pi : \mathcal{A} \times \mathcal{S} \times \mathcal{K} \rightarrow \mathbb{R}$, $a_t \sim \pi(a_t \mid s_t, k)$ aims to steer the environment into a target state $s_T$ for which it holds that $\Phi(s_T) = \hat{z}$, given that $\Phi(s_0) = z$ and $\hat{z} = \text{argmax}_{z'} q_\theta(z' \mid z, k)$ (see Fig. 2). We can relate this discretely indexed family of policies to a set of $K$ *options* [20]. An option is formally defined as a triple $o_k = (I_k, \beta_k, \pi_k)$ where $I_k \subseteq \mathcal{S}$ is the set of states in which option $k$ is applicable, $\beta_k : \mathcal{S} \times \mathcal{S} \rightarrow [0, 1]$, $\beta_k(s_0, s_t)$ parametrizes a Bernoulli probability of termination in state $s_t$ when starting in $s_0$ (f.e. when detecting an abstract state change) and $\pi_k(a \mid s_t) : \mathcal{S} \rightarrow \Delta(\mathcal{A})$ is the option policy on the action space $\mathcal{A}$. We will refer to the option policy as *skill policy* in the following. We assume that all options are applicable in all states, i.e., $I_k = \mathcal{S}$. An option terminates if the symbolic state has changed between $s_0$ and $s_t$ or a timeout is reached, i.e., $\beta_k(s_0, s_t) = \mathbf{1}[\Phi(s_0) \neq \Phi(s_t) \vee t = t_{\max}]$. To this end, we append a normalized counter $t/t_{\max}$ to the state $s_t$. We define the operator apply as $s_T = \text{apply}(E, \pi, s_0, k)$ which applies the skill policy $\pi(a_t \mid s_t, k)$ until termination on environment $E$ starting from initial state $s_0$ and returns the terminal state $s_T$. We also introduce a bracketed time notation which abstracts the effect of skill execution from the number of steps $T$ taken until termination $s_{(n)} = \text{apply}(E, \pi, s_{(n-1)}, k)$ with $n \in \mathbb{N}_0$ (see Fig. 2a). The apply operator can thus be rewritten as $s_{(1)} = \text{apply}(E, \pi, s_{(0)}, k)$ with $s_{(0)} = s_0$, $s_{(1)} = s_T$. The symbolic forward model $q_\theta(z_T \mid z_0, k)$ aims to capture the relation of $z_0$, $k$ and $z_T$ for $s_T = \text{apply}(E, \pi, s_0, k)$ with $z_0 = \Phi(s_0), z_T = \Phi(s_T)$. The model factorizes over the symbolic observation as $q_\theta(z_T \mid k, z_0) = \prod_{d=1}^D q_\theta([z_T]_d \mid k, z_0) = \prod_{d=1}^D \text{Bernoulli}([\alpha_T]_d)$. The Bernoulli probabilities $\alpha_T : \mathcal{Z} \times \mathcal{K} \rightarrow (0, 1)^D$ are predicted by a neural component. We use a multilayer perceptron (MLP) $f_\theta$ which predicts the probability $p_{\text{flip}}$ of binary variables in $z_0$ to toggle $p_{\text{flip}} = f_\theta(z_0, k)$. The index operator $[x]_d$ returns the d$^{\text{th}}$ element of vector $x$.

**Objective**  For any state $s_0 \in \mathcal{S}$ with associated symbolic state $z_0 = \Phi(s_0)$ we aim to learn $K$ skills $\pi(a \mid s_t, k)$ which maximize the diversity in the set of reachable successor states $\{z_T^k = \Phi(\text{apply}(E, \pi, s_0, k)) \mid k \in \mathcal{K}\}$. Jointly, we aim to model the effect of skill execution with the forward model $q_\theta(z_T \mid z_0, k)$. Inspired by Variational Intrinsic Control [10] we take an information-theoretic perspective and maximize the mutual information $\mathcal{I}(z_T, k \mid z_0)$ between the skill index $k$ and the symbolic observation $z_T$ at skill termination given the symbolic observation $z_0$ at skill initiation, i.e., $\max \mathcal{I}(z_T, k \mid z_0) = \max H(z_T \mid z_0) - H(z_T \mid z_0, k)$. The intuition behind this objective function is that we encourage the agent to (i) reach a diverse set of terminal observations $z_T$ from an initial observation $z_0$ (by maximizing the conditional entropy $H(z_T \mid z_0)$) and (ii) behave predictably such that the terminal observation $z_T$ is ideally fully determined by the initial observation $z_0$ and skill

index $k$ (by minimizing $H(z_T \mid z_0, k)$). We reformulate the objective as an expectation over tuples $(s_0, k, s_T)$ by employing the mapping function $\Phi$ as $\mathcal{I}(z_T, k \mid z_0) = \mathbb{E}_{(s_0, k, s_T) \sim P} \left[ \log \frac{p(z_T \mid z_0, k)}{p(z_T \mid z_0)} \right]$ with $z_T := \Phi(s_T), z_0 := \Phi(s_0)$ and replay buffer $P$. Similar to [12] we derive a lower bound on the mutual information, which is maximized through the interplay of a RL problem and maximum likelihood estimation. To this end, we first introduce a variational approximation $q_\theta(z_T \mid z_0, k)$ to the transition probability $p(z_T \mid z_0, k)$, which we model by a neural component. We decompose

$$\mathcal{I}(z_T, k \mid z_0) = \mathbb{E}_{(s_0, k, s_T) \sim P} \left[ \log \frac{q_\theta(z_T \mid z_0, k)}{p(z_T \mid z_0)} \right] + \underbrace{\mathbb{E}_{(s_0, k, s_T) \sim P} \left[ \log \frac{p(z_T \mid z_0, k)}{q_\theta(z_T \mid z_0, k))} \right]}_{\approx D_{\mathrm{KL}}(p(z_T \mid z_0, k) \,||\, q_\theta(z_T \mid z_0, k))}$$

giving rise to the lower bound $\mathcal{I}(z_T, k \mid z_0) \geq \mathbb{E}_{(s_0, k, s_T) \sim P} \left[ \log \frac{q_\theta(z_T \mid z_0, k)}{p(z_T \mid z_0)} \right]$ whose maximization can be interpreted as a sparse-reward RL problem with reward $\hat{R}_T(k) = \log \frac{q_\theta(z_T \mid z_0, k)}{p(z_T \mid z_0)}$. We approximate $p(z_T \mid z_0)$ as $p(z_T \mid z_0) \approx \sum_{k'} q_\theta(z_T \mid z_0, k') p(k' \mid z_0)$ and assume $k$ uniformly distributed and independent of $z_0$, i.e. $p(k' \mid z_0) = \frac{1}{K}$. This yields a tractable reward

$$R_T(k) = \log \frac{q_\theta(z_T \mid z_0, k)}{\sum_{k'} q_\theta(z_T \mid z_0, k')} + \log K. \tag{1}$$

In Sec. 3 we describe modifications we apply to the intrinsic reward $R_T$ which improve the performance of our proposed algorithm. To tighten the lower bound, the KL divergence term in eq. (1) has to be minimized. Minimizing the KL divergence term corresponds to "training" the symbolic forward model $q_\theta$ by maximum likelihood estimation of the parameters $\theta$.

**Training procedure**  In each epoch of training, we first collect skill trajectories on the environment using the skill policy $\pi$. For each episode $i \in \{1, ..., N\}$ we reset the environment and obtain an initial state $s_0^i$. Next, we uniformly sample skills $k^i \sim \mathrm{Uniform}\{1, ..., K\}$. By iteratively applying the skill policy $\pi(\cdot \mid \cdot, k^i)$ we obtain resulting states $s_0^i...s_{T_i}^i$ and actions $a_0^i...a_{T_i-1}^i$. A skill rollout terminates either if an environment-dependent step-limit is reached or when a change in the symbolic observation $z_t \neq z_0$ is observed. We append the rollouts to a limited-size buffer $\mathcal{B}$. In each training epoch we sample two sets of episodes $\mathcal{S}_{\mathrm{RL}}, \mathcal{S}_{\mathrm{FM}}$ from the buffer for training the policy $\pi$ and symbolic forward model $q_\theta$. Both episode sets are relabelled as described in Sec. 3. Let $i \in \{1, ..., M\}$ now refer to the episode index in the set $\mathcal{S}_{\mathrm{RL}}$. From $\mathcal{S}_{\mathrm{RL}}$ we sample transition tuples $([s_t^i, k^i], [s_{t+1}^i, k^i], a_t^i, r_{t+1}^i(k^i))$ which are used to update the policy $\pi$ using the soft actor-critic (SAC) algorithm [38]. To condition the policy on skill $k$ we concatenate $k$ to the state $s$ as denoted by $[\cdot, \cdot]$. We set the intrinsic reward to zero $r_{t+1}^i = 0$ except for the last transition in an episode ($t + 1 = T_i$) in which $r_{t+1}^i(k^i) = R(k^i)$. From the episodes in $\mathcal{S}_{\mathrm{FM}}$ we form tuples $(z_0^i = \Phi(s_0^i), k^i, z_{T_i}^i = \Phi(s_{T_i}^i))$ which are used to train the symbolic forward model using gradient descent.

**Relabelling**  Early in training, the symbolic transitions caused by skill executions mismatch the predictions of the symbolic forward model. We can in *hindsight* increase the match between skill transitions and forward model by replacing the actual $k^i$ which was used to collect the episode $i$ by a different $k_*^i$ in all transition tuples $([s_t^i, k_*^i], [s_{t+1}^i, k_*^i], a_t^i, r_{t+1}^i(k_*^i))$ and $(z_0^i, k_*^i, z_{T_i}^i)$ of episode $i$. In particular, we aim to replace $k^i$ by $k_*^i$ which has highest probability $k_*^i = \max_k q_\theta(k \mid z_T, z_0)$. However, this may lead to an unbalanced distribution over $k_*^i$ after relabelling which is no longer uniform. To this end, we introduce a constrained relabelling scheme as follows. We consider a set of episodes indexed by $i \in \{1, ..., N\}$ and compute skill log-probabilities for each episode which we denote by $Q_k^i = \log q_\theta(k \mid z_0^i, z_{T_i}^i)$ where $q_\theta(k \mid z_0^i, z_{T_i}^i) = \frac{q_\theta(z_{T_i}^i \mid z_0^i, k)}{\sum_{k'} q_\theta(z_{T_i}^i \mid z_0^i, k')}$. We find a relabeled skill for each episode $(k_*^1, ..., k_*^N)$ which maximizes the scoring $\max_{(k_*^1, ..., k_*^N)} \sum_i Q_{k_*^i}^i$ under the constraint that the counts of re-assigned skills $(k_*^1, ..., k_*^N)$ and original skills $(k^1, ..., k^N)$ match, i.e. $\#_{i=1}^N [k_*^i = k] = \#_{i=1}^N [k^i = k] \quad \forall k \in \{1, ..., K\}$ which is to ensure that after relabelling no skill is over- or underrepresented. The count operator $\#[\cdot]$ counts the number of positive (true) evaluations of its argument in square brackets. This problem can be formulated as a linear sum assignment problem which we solve using the Hungarian method [39, 40]. While we pass all episodes in $\mathcal{S}_{\mathrm{FM}}$ to the relabelling module, only a subset (50%) of episodes in $\mathcal{S}_{\mathrm{RL}}$ can potentially be relabeled to retain negative examples for the SAC agent. Relabelling experience in hindsight to improve sample efficiency is a common approach in goal-conditioned [41] and hierarchical [19] RL.

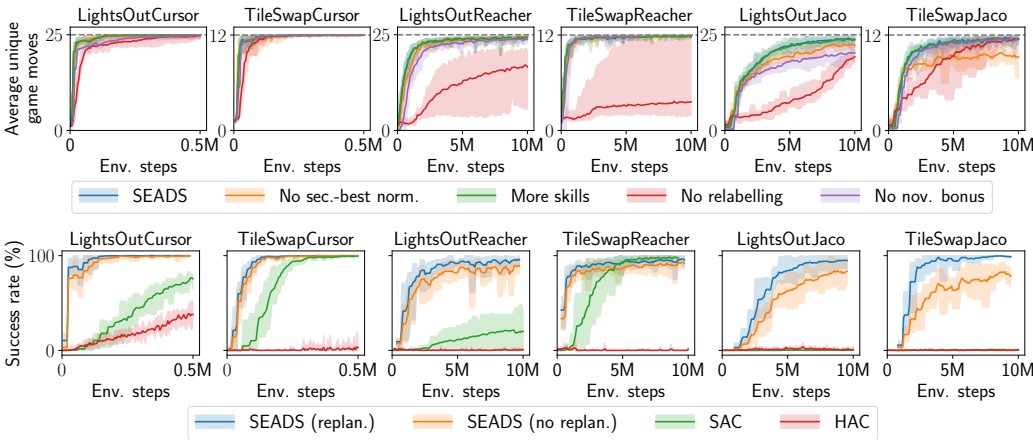

Figure 3: Top row: Number of learned unique game moves for ablations of SEADS. Bottom row: Success rate of the proposed SEADS agent and baseline methods on *LightsOut*, *TileSwap* games embedded in *Cursor*, *Reacher*, *Jaco* environments. SEADS performs comparably or outperforms the baselines on all tasks. The solid line depicts the mean, shaded area min. and max. of 10 (SEADS, SAC on *Cursor*) / 5 (HAC, SAC on *Reacher*, *Jaco*) independently trained agents.

**Reward improvements**    The reward in eq. (1) can be denoted as $R(k) = \log q_\theta(k \mid z_0, z_T) + \log K$. For numerical stability, we define a lower bounded term $\bar{Q}_k = \mathrm{clip}(\log q_\theta(k \mid z_0, z_T), \min = -2\log(K))$ and write $R^0(k) = \bar{Q}_k + \log K$. In our experiments, we observed that occasionally the agent is stuck in a local minimum in which (i) the learned skills are not unique, i.e., two or more skills $k \in \mathcal{K}$ cause the same symbolic transition $z_0 \to z_T$. In addition, (ii), occasionally, not all possible symbolic transitions are discovered by the agent. To tackle (i) we reinforce the policy $\pi$ with a positive reward if and only if no other skill $k'$ better fits the symbolic transition $(z_0 \to z_T)$ generated by $\mathrm{apply}(E, \pi, s_0, k)$, i.e., $R^{\mathrm{norm}}(k) = \bar{Q}_k - \mathrm{top2}_{k'} \bar{Q}_{k'}$, which we call **second-best normalization**. The operator $\mathrm{top2}_{k'}$ selects the second-highest value of its argument for $k' \in \mathcal{K}$. We define $R^{\mathrm{base}}(k) = R^{\mathrm{norm}}(k)$ except for the "No second-best norm." ablation where $R^{\mathrm{base}}(k) = R^0(k)$. To improve (ii) the agent obtains a **novelty bonus** for transitions $(z_0 \to z_T)$ which are not modeled by the symbolic forward model for *any* $k'$ by $R(k) = R^{\mathrm{base}}(k) - \max_{k'} \log q_\theta(z_T \mid z_0, k')$. If the symbolic state does not change ($z_T = z_0$), we set $R(k) = -2\log(K)$ (minimal attainable reward).

**Planning and skill execution**    A task is presented to our agent as an initial state of the environment $s_0$ with associated symbolic observation $z_0$ and a symbolic goal $z^*$. First, we leverage our learned symbolic forward model $q_\theta$ to plan a sequence of skills $k_1, ..., k_N$ from $z_0$ to $z^*$ using breadth-first search (BFS). We use the mode of the distribution over $z'$ for node expansion in BFS: $\mathrm{successor}_{q_\theta}(z, k) = \mathrm{argmax}_{z' \in \mathcal{Z}} q_\theta(z' \mid z, k)$. After planning, the sequence of skills $[k_1, ..., k_N]$ is iteratively applied to the environment through $s_{(n)} = \mathrm{apply}(E, \pi, s_{(n-1)}, k_n)$. Inaccuracies of skill execution (leading to different symbolic observations than predicted) can be coped with by replanning after each skill execution. Both single-outcome (mode) determinisation and replanning are common approaches to probabilistic planning [42]. We provide further details in the supplementary material.

## 4   Experiments

We evaluate our proposed agent on a set of physically-embedded game environments. We follow ideas from [5] but consider single-player games which in principle enable full control over the environment without the existence of an opponent. We chose *LightsOut* and *TileSwap* as board games which are embedded in a physical manipulation scenario with *Cursor*, *Reacher* or *Jaco* manipulators (see Fig. 1). The *LightsOut* game (see Figure 1) consists of a $5 \times 5$ board of fields. Each field has a binary illumination state of *on* or *off*. By pushing a field, its illumination state and the state of the (non-diagonally) adjacent fields toggles. At the beginning of the game, the player is presented a board where some fields are on and the others are off. The task of the player is to determine a set of fields to push to obtain a board where all fields are off. The symbolic observation in all *LightsOut* environments represents the illumination state of all 25 fields on the board $\mathcal{Z} = \{0, 1\}^{5 \times 5}$.

In *TileSwap* (see Fig. 1) a $3 \times 3$ board is covered by chips numbered from 0 to 8 (each field contains exactly one chip). Initially, the chips are randomly assigned to fields. Two chips can be swapped if they are placed on (non-diagonally) adjacent fields. The game is successfully finished after a number of swap operations if the chips are placed on the board in ascending order. In all *TileSwap* environments the symbolic observation represents whether the i-th chip is located on the j-th field $\mathcal{Z} = \{0, 1\}^{9 \times 9}$. To ensure feasibility, we apply a number of random moves (pushes/swaps) to the goal board configuration of the respective game. We quantify the difficulty of a particular board configuration by the number of moves required to solve the game. We ensure disjointness of board configurations used for training and testing through a hashing algorithm (see supp. material). A board game move ("push" in *LightsOut*, "swap" in *TileSwap*) is triggered by the manipulator's end effector touching a particular position on the board. We use three manipulators of different complexity (see Fig. 1). The *Cursor* manipulator can be navigated on the 2D game board by commanding $x$ and $y$ displacements. The board coordinates are $x, y \in [0, 1]$, the maximum displacement per timestep is $\Delta x, \Delta y = 0.2$. A third action triggers a push (*LightsOut*) or swap (*TileSwap*) at the current position of the cursor. The *Reacher* [1] manipulator consists of a two-link arm with two rotary joints. The position of the end effector in the 2D plane can be controlled by external torques applied to the two rotary joints. As for the *Cursor* manipulator an additional action triggers a game move at the current end effector coordinates. The *Jaco* manipulator [43] is a 9-DoF robotic arm whose joints are velocity-controlled at 10Hz. It has an end-effector with three "fingers" which can touch the underlying board to trigger game moves. The arm is reset to a random configuration above the board around the board's center after a game move (details in supplementary material). By combining the games of *LightsOut* and *TileSwap* with the *Cursor*, *Reacher* and *Jaco* manipulators we obtain six environments. For the step limit for skill execution we set 10 steps on *Cursor* and 50 steps in *Reacher* and *Jaco* environments. With our experiments we aim at answering the following research questions: **R1:** How many distinct skills are learned by SEADS? Does SEADS learn all 25 (12) possible moves in *LightsOut* (*TileSwap*) reliably? **R2:** How do our design choices contribute to the performance of SEADS? **R3:** How well does SEADS perform in solving the posed tasks in comparison to other flat and hierarchical RL approaches? **R4:** Can our SEADS also be trained and applied on a real robot?

**Skill learning evaluation**    To address **R1** we investigate how many distinct skills are learned by SEADS. If not all possible moves within the board games are learned as skills (25 for *LightsOut*, 12 for *TileSwap*), some initial configurations can become unsolvable for the agent, negatively impacting task performance. To count the number of learned skills we apply each skill $k \in \{1, ..., K\}$ on a fixed initial state $s_0$ of the environment $E$ until termination (i.e., $\text{apply}(E, s_0, \pi, k)$). Among these $K$ skill executions we count the number of unique game moves being triggered. We report the average number of unique game moves for $N = 100$ distinct initial states $s_0$. On the *Cursor* environments SEADS detects nearly all possible game moves (avg. approx. 24.9 of 25 possible in LightsOutCursor, 12 of 12 in TileSwapCursor). For *Reacher* almost all moves are found (24.3/11.8). In the *Jaco* environments some moves are missing occasionally (23.6/11.5). We demonstrate superior performance compared to a baseline skill discovery method (Variational Intrinsic Control, Gregor et al. [10]) in the supplementary material. We substantiate our agent design decisions through an ablation study (**R2**) in which we compare the number of unique skills (game moves) detected for several variants of SEADS (see Fig. 3). In a first study, we remove single parts from our agent to quantify their impact on performance. This includes training SEADS without the proposed *relabelling*, *second-best normalization* and *novelty bonus*. We found all of these innovations to be important for the performance of SEADS, with the difference to the full SEADS agent being most prominent in the LightsOutJaco environment. Learning with *more skills* (15 for *TileSwap*, 30 for *LightsOut*) than actually needed does not harm performance.

**Task performance evaluation**    To evaluate the task performance of our agent and baseline agents (**R3**) we initialize the environments such that the underlying board game requires at maximum 5 moves (pushes in *LightsOut*, swaps in *TileSwap*, limited due to branching factor and BFS) to be solved. We evaluate each agent on 20 examples for each number of moves in $\{1, ..., 5\}$ required to solve the game. We consider a task to be successfully solved if the target board configuration was reached (all fields *off* in *LightsOut*, ordered field in *TileSwap*). For SEADS we additionally count tasks as "failed" if planning exceeds a wall time limit of 60 seconds. We evaluate both planning variants with and without replanning. As an instance of a flat (non-hierarchical) agent we evaluate the performance of Soft Actor-Critic (SAC [38]). The SAC agent receives the full environment state $s \in \mathcal{S}$ which includes the symbolic observation (board state). It obtains a reward of 1 if it successfully solved the game and 0 otherwise.

In contrast to the Soft Actor-Critic agent the SEADS agent leverages the decomposition of state $s \in \mathcal{S}$ and symbolic observation $z \in \mathcal{Z}$. For a fair comparison to a hierarchical agent, we consider Hierarchical Actor-Critic (HAC, Levy et al. [19]), which, similar to SEADS, can also leverage the decomposition of $s$ and $z$. We employ a two-level hierarchy in which the high-level policy sets *symbolic* subgoals $z \in \mathcal{Z}$ to the low-level policy, thereby leveraging the access to the symbolic observation. We refer to the supplementary material for implementation and training details of SAC and HAC. Fig. 3 visualizes the performance of SEADS and the baselines. On all environments SEADS performs similar or outperforms the baselines, with the performance difference being most pronounced on the *Jaco* environments on which SAC and HAC do not make any progress. On the *Cursor* environments SEADS achieves a success rate of 100% after $5 \cdot 10^5$ environment steps. On the remaining environments, the average success rate (with replanning) is 95.8% (LightsOutReacher), 95.5% (TileSwapReacher), 94.9% (LightsOutJaco), 98.8% (TileSwapJaco) after $10^7$ steps.

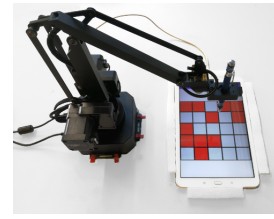

Figure 4: Robot setup.

**Robot experiment**    To evaluate the applicability of our approach on a real-world robotic system (**R4**) we set up a testbed with a uArm Swift Pro robotic arm which interacts with a tablet using a capacitive pen (see Fig. 4). The SEADS agent commands a displacement $|\Delta x|, |\Delta y| \leq 0.2$ and an optional pushing command as in the *Cursor* environments. The board state is communicated to the agent through the tablet's USB interface. We manually reset the board *once* at the beginning of training, and do not interfere in the further process. After training for $\approx 160k$ interactions ($\approx 43.5$ hours) the agent successfully solves all boards in a test set of 25 board configurations (5 per solution depth in $\{1, ..., 5\}$). We refer to the supplementary material for details and a video.

## 5    Assumptions and Limitations

Our approach assumes that the state abstraction is known, the symbolic observation $z$ is provided by the environment, and that the continuous state is fully observable. Learning the state abstraction too is an interesting direction for future research. The breadth-first search planner we use for planning on the symbolic level exhibits scaling issues for large solution depths; e.g., for LightsOut it exceeds a 5-minute threshold for solution depths (number of initial board perturbations) $\geq 9$. In future work, more efficient heuristic or probabilistic planners could be explored. Currently, our BFS planner produces plans which are optimal with respect to the number of skills executed. Means for predicting and taking the skill execution cost into account for planning could be pursued in future work. In the more complex environments (*Reacher*, *Jaco*) we observe our agent to not learn all possible skills reliably, in particular for skills for which no transitions exist in the replay buffer. In future work one could integrate additional exploration objectives which incentivize to visit unseen regions of the state space. Also, the approach is currently limited to settings such as in board games, where all symbolic state transitions should be mapped to skills. It is an open research question how our skill exploration objective could be combined with demonstrations or task-specific objectives to guide the search for symbolic actions and limit the search space in more complex environments.

## 6    Conclusion

We present an agent which, in an unsupervised way, learns diverse skills in complex physically embedded board game environments which relate to *moves* in the particular games. We assume a state abstraction from continuous states to symbolic states known and observable to the agent as prior information, and that skills lead to changes in the symbolic state. The jointly learned forward model captures the temporally extended *effects* of skill execution. We leverage this forward model to plan over a sequence of skills (moves) to solve a particular task, i.e., bring the game board to a desired state. We demonstrate that with this formulation we can solve complex physically embedded games with high success rate, that our approach compares favorably with other flat and hierarchical RL algorithms, and also transfers to a real robot. Our approach provides an unsupervised learning alternative to prescribing the action abstraction and pretraining each skill individually before learning a forward model from skill executions. In future research, our approach could be combined with state abstraction learning to leverage its full potential.

**Acknowledgments**

This work was supported by Max Planck Society and Cyber Valley. The authors thank the International Max Planck Research School for Intelligent Systems (IMPRS-IS) for supporting Jan Achterhold.

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
