# OpenReview forum: "Learning Temporally Extended Skills in Continuous Domains as Symbolic Actions for Planning"
_robot-learning.org/CoRL/2022/Conference — CoRL 2022 Oral_

### Official Review · Reviewer_XcfM · 2022-07-04

**Originality:** Good
**Technical Quality:** Very Good
**Clarity Of Presentation:** Very Good
**Impact:** 2

**Recommendation:**

Weak Accept: I recommend accepting the paper, but will not argue for my recommendation if the majority of other reviewers have a different opinion.

**Summary:**

This work proposes to use a given symbolic state abstraction to learn and plan with temporally extended skills. The skills are learned by reinforcement learning, with an objective that combines diversity and predictability in terms of reachable symbolic states. Additional tricks are proposed to help avoid local minima. Experiments are in physically embodied grid-based games.


**Issues:**

* A “more skills” variation of the main method is briefly mentioned in S.5, with results in Figure S.4, but the variation is never explained (e.g., how many more skills are there?) I was going to ask to see results for a variation like this, so I am interested to know the details.
* It’s not clear why that “more skills” variation and the VIC results are in the appendix, rather than in the main paper. (Maybe they were generated in between the paper deadline and the supplementary deadline?)
* Other references:
   * “Leveraging Approximate Symbolic Models for Reinforcement Learning via Skill Diversity” Guan et al. 2022
   * “Reprel: Integrating relational planning and reinforcement learning for effective abstraction” Kokel et al. 2021
   * “Symbolic Plans as High-Level Instructions for Reinforcement Learning” Illanes et al. 2020
   * “SDRL: interpretable and data-efficient deep reinforcement learning leveraging symbolic planning” Lyu et al. 2019
   * “Using Abstract Models of Behaviours to Automatically Generate Reinforcement Learning Hierarchies” Ryan 2002
   * “Integrated Task and Motion Planning” Garrett et al. 2021

### Minor
* $\mathcal{S}_0$ is not defined on L139, and it seems like just $\mathcal{S}$ is used instead starting on L143.
* It’s not immediately obvious why $I$ and $\beta$ are defined over tuples of states rather than individual states. In the original options paper, they’re just defined over individual states. My understanding is that they are defined in this way because the termination function needs to “remember” the original abstract state so that it can terminate when the abstract state changes. It would be helpful to spell this out.
* L141 grammar: “for the option policy holds”
* L218: remove period before “which”
* L246 grammar: “As step limit for skill execution”
* There are a number of places where the open quotation marks are backwards



**Quality Of The Limitations Section:**

Limitations are addressed clearly

**Reviewer Expertise:**

4: The reviewer is confident but not absolutely certain that the evaluation is correct

**Robotics Focus:**

Sufficient demonstration on hardware

**Strengths And Weaknesses:**

### Strengths

* The writing is overall very clear. I did not find many grammatical issues and the organization is good.
* It is clear how the proposed approach improves on less structured and more model-free approaches like SAC, VIC, and HAC, and it is good to see those baselines in the experiments.
* The ablations in Figure 3 and the appendix are very helpful for understanding the relative contributions of the proposed approach.
* It’s always impressive to see results with reinforcement learning on a real robot, and the video does a good job visualizing that process.
* The discussion of limitations is helpful and thorough.


### Weaknesses

* The environments used in the experiments are a bit contrived, and it is unclear to what extent the approach proposed in this paper would work beyond these environments.
   * It is unclear whether the robot is really adding anything in these environments. In other words, one could argue that the robot is added just to make the demos more flashy, when in reality, all of the technical difficulty resides in the original board games.
   * If these environments were actually ones that we wish to solve in a real application, there would be much easier ways to solve them, and those ways would probably not involve machine learning. For example, use a symbolic planner to solve the game boards, and use inverse kinematics and a hardcoded press-down motion for the temporally extended actions. (This would be a simple version of task and motion planning, but where the task and motion planning aspects are completely decoupled.)
   * Given that the symbolic state space is highly structured, and it is known what the ideal abstract actions would be (pressing each grid cell), a much more direct RL approach would (1) create one skill for each grid cell; and (2) directly associate skills to cells, and give rewards accordingly.
* The fact that the number of skills needs to be manually specified is a big limitation, and in experiments, the direct correspondence between the number of skills and the symbolic action space is a big injection of knowledge that is not emphasized.
* There are several related works in the AI planning community (see Issues below) that call into question the novelty of this work, but this paper is the first that I have seen with (real) robotics results.
* If the objective of this work is to learn temporally extended skills, the skills learned in the experiments are all essentially the same at the policy level: move and press. It would be more interesting to see a variety of skills learned.
* I’m not completely convinced that the “second-best normalization” idea makes sense. Intuitively, it seems redundant with the diversity objective, but I can understand how optimizing that objective directly could lead to local minima. Moreover, the empirical results do not show a significant improvement vs. the “no sec-best norm” ablation.
* It is not clear to what extent generalization is evaluated in the experiments. L250 states that board configurations are disjoint in training and testing, which is a good start, but I would be curious to know: in the course of a testing episode, how many of the symbolic states encountered are novel with respect to the training distribution? In addition to measuring this directly, it would be informative to see an ablation where the learned forward model is replaced with a tabular model.
* As the paper mentions, planning with breadth first search is very limiting. Given the symbolic states and the existence of powerful AI planners, it would be more impressive if the overall approach were able to use those planners. For example, the works by Asai et al. and Konidaris et al. use PDDL planners and can scale to much longer horizons than the ones in this work.


**Summary Of Recommendation:**

*Post-rebuttal:* Thanks to the authors for addressing my questions and concerns. I am raising my recommendation to Weak Accept.

*Pre-rebuttal:*
The work is overall performed with a high level of technical sophistication, and the paper is well-written, but I am concerned about novelty and about the limitations of the experimental environments.

---

> ### Author Response · Authors · 2022-08-26
> **Author response: Part 2**
>
> ### Learning with a tabular model
> Thank you for the interesting suggestion to use a tabular model to model the effect of skill executions $z, k \rightarrow \hat{z}$ on the symbolic level. However, we see a significant issue when memorizing observed transitions $(z, k, \hat{z})$ with a tabular model, which is the distribution shift on the transition dynamics on the symbolic level induced by changes in the low-level skill policies. When updating the skill policies $\pi$ during training, old transitions $(z, k, \hat{z})$ can become invalid (due to updates on the policies, skill $k$ executed on state $z$ may lead to a symbolic state $\hat{z}^* \neq \hat{z})$. Such a distribution shift will happen inevitably: At the beginning of training, the skill policies are basically executing random actions, while at the end of training, skills $k$ should relate to diverse transitions in the symbolic state space. Furthermore, old transitions in the tabular model can not easily be corrected: There is no "model" which would allow to simulate the effect of skill $k$ on state $z$.
>
> One approach to the distribution shift problem would be to only memorize transitions observed for recent policy executions. This, however, contradicts with the requirement that many transitions $(z, k, \hat{z})$ are required for a planner to traverse the symbolic state space to find a sequence of skills leading from an initial state to a target state.
>
> In contrast to the tabular model, a parametrized model such as a neural network can be incrementally updated to reflect the changes in the effect of executing a low-level policy. Furthermore, it allows to *generalize* to symbolic states which were not seen during training (see "Overlap of training and testing during task execution").
>
>
> ### BFS is limiting the applicability of the method / PDDL representation
> We agree that extracting PDDL representations as forward models is an interesting avenue of future work.
> For our board games, the high branching factor of the planning problem and the use of plain BFS without heuristics/pruning is currently limiting the planning performance.
> Note that by PDDL itself, the complexity of the planning problem is not changed, but as suggested by the reviewer more efficient planners that use heuristics and pruning strategies could be pursued in future work.
> Please also refer to sec. S.5 in the updated supplementary material for an analysis of BFS planning times and success rates for different solution depths (up to depth 9) on the LightsOutCursor environment.
>
> ### Issue: Explain "more skills" ablation
> Thanks for pointing this out, we added the missing details to the paper and supplementary material.
>
> ### Why are "more skills" and VIC in the appendix
> We now added the more skills ablation to the main paper.
> The comparison with VIC is provided as an additional supplemental result in the supplementary material.
> Due to space limitations the figures unfortunately would not fit into the main paper.
>
> ### Other references
>
> Thank you for pointing us to further related work. We will include them in the paper.
> - “Leveraging Approximate Symbolic Models for Reinforcement Learning via Skill Diversity” Guan et al. 2022
> - “Reprel: Integrating relational planning and reinforcement learning for effective abstraction” Kokel et al. 2021
> - “Symbolic Plans as High-Level Instructions for Reinforcement Learning” Illanes et al. 2020
> - “SDRL: interpretable and data-efficient deep reinforcement learning leveraging symbolic planning” Lyu et al. 2019
> - “Using Abstract Models of Behaviours to Automatically Generate Reinforcement Learning Hierarchies” Ryan 2002
>
> The above approaches combine symbolic planning in a given or extracted domain description (state and action abstractions) with reinforcement learning to execute the symbolic actions.
> Similar to our approach, the work in Guan et al. learns low-level policies using an information-theoretic skill diversity measure which implement the symbolic actions.
> The approach extracts landmark states from given symbolic state and action representations as subgoals for skill learning, while our approach learns the action abstraction with associated skill policies given the state abstraction.
>
> - “Integrated Task and Motion Planning” Garrett et al. 2021
>
> Surveys task and motion planning approaches which use predefined state and action abstractions for planning on a symbolic level, assuming that the scene can be sufficiently well be modeled for kinematic motion planning to implement the symbolic actions.
> Learning of action abstractions and skill execution is not pursued in these methods.
>
> ### Minor comments
> Fixed, thank you.

---

> ### Author Response · Authors · 2022-08-26
> **Author response: Part 1**
>
> We thank the reviewer for the time and the valuable suggestions to improve our paper.
> In the following, we answer the questions raised, and detail how we updated our paper based on the suggestions.
>
> ### Unclear how the approach scales beyond board game environments
> To transfer the algorithm to more complex environments such as object stacking tasks, more research is necessary.
> We expect that improvements for the exploration of low-level skills would be needed instead of random action perturbations in SAC which give incentives to explore unknown regions of the low-level state space.
> Also adding demonstrations could help to confine the search space.
>
> ### Why physically embedded games with robots
>
> We respectfully disagree that "all of the technical difficulty resides in the original board games".
> Our approach is about the technical challenge of learning action abstractions with diverse and predictable skills that link symbolic state changes with low-level control policies.
> The various environment types have different levels of complexity by varying degrees of freedom of the agent that need to be controlled.
> A forward model on the symbolic level is concurrently learned which predicts the effect of skill execution on the symbolic states and allows for planning in the symbolic domain.
> As we demonstrate in our ablations, this learning problem is not trivial and requires careful design of the learning algorithm.
>
>
> ### Our approach vs are more direct approach
> The board games could be solved with more manual engineering too (providing the action abstraction), however, we are interested in self-supervised learning methods that learn action abstraction from interactions with the environment given the state abstraction.
>
>
> ### Number of skills must be pre-specified
> Please observe the "more skills" ablation which has been moved from the supplementary to the main paper. Skill/action abstraction learning also works if the number of skills is larger than needed.
>
>
> ### All skills are "move and press"
> We agree that the skills exhibits semantically similar behaviour. However, we want to stress that in the Jaco environments, the skills do not follow the "move to location in 2D and execute push" scheme from the Cursor and Reacher environments. Rather, more complex behavior is learned, featuring low-level control of the robotic arm and end-effector in 3D, and executing moves on the board games through pushes with the end-effectors' fingers.
>
> ### Second-best norm. does not give a significant improvement
> Please note the significant improvement on the more complex Jaco environments in Table 1 of the supplementary material and the faster convergence for skill learning in Fig. 3 in the main paper when using the second-best normalization.
>
> ### Generalization/overlap of training and testing board configurations during task execution
> To assess how many board configurations appear during solving particular LightsOut and TileSwap instances, we recorded all boards which appear when training LightsOutCursor and TileSwapCursor on 500k environment interactions. We trained 10 SEADS agent on each environment independently. For each of those agents, we then solved 5 problem instances for solution depths {1, ... 5} from the test set of the respective environments. During solving these problems, we recorded how many board configurations on the "path" from the initial board configuration to the target board configuration (all  lights off in LightsOut, ordered fields in TileSwap) were seen during training. We found that a majority of boards occuring during task evaluation do *not* appear during training (LightsOut: $19.7\% \pm 3.8\%$ seen, TileSwap: $32.6\% \pm 1.6\%$ seen).
> We would like to stress that the initial board configurations used for task evaluation were never seen during training due to the train/test split separation.

---

### Official Review · Reviewer_a18m · 2022-07-26

**Originality:** Good
**Technical Quality:** Very Good
**Clarity Of Presentation:** Good
**Impact:** 3

**Recommendation:**

Weak Accept: I recommend accepting the paper, but will not argue for my recommendation if the majority of other reviewers have a different opinion.

**Summary:**

This paper proposes ‘SEADS’, a novel hierarchical approach to learning diverse skills as well as a forward model over these skills that will be useful for planning to solve long-horizon tasks. The proposed method assumes that a discrete, symbolic abstraction of the low-level continuous environment is provided as input. Given this, it utilizes an unsupervised exploration reward to collect data via environment interaction, and then learns skills and the corresponding forward model based on these data. Experiments show that the proposed approach is capable of solving two different physically-embedded games (LightsOut and TileSwap) under different settings more efficiently than other baseline hierarchical RL approaches.

**Issues:**

- As mentioned in the “weaknesses” section, I find the approach’s applicability to be rather limited to physical games such as LightsOut and TileSwap. Do the authors agree with this assessment? If so, I think it is important to mention this in the Limitations section. Additionally, I think it is important to highlight the assumptions regarding the size of the state space and desirability of reaching all of it in Section 3.
- As mentioned in the “weaknesses” section, I don’t understand why the evaluation was restricted to tasks that require only up to 5 moves. I think it is important to justify this choice in the text.
- When reading through the paper, it took me a while to understand that the number *K* is being provided as input to the approach. I think it’s important to mention this explicitly at the start of Section 3.

**Quality Of The Limitations Section:**

Additional details required

**Reviewer Expertise:**

4: The reviewer is confident but not absolutely certain that the evaluation is correct

**Robotics Focus:**

Sufficient demonstration on hardware

**Strengths And Weaknesses:**

**Strengths**

- The proposed method seems novel and is intuitive. I thought the overall idea to learn skills and the forward model jointly was interesting. Moreover, the derivation of the exploration reward approximation, as well as the relabelling were both clever and creative ideas.
- The paper is generally well-written, clear and easy-to-follow.

**Weaknesses**

- The proposed approach applies only to a rather limited setting. To me, it seems that a key limitation of this work is the set of assumptions that (1) the abstract symbolic state space is small, and (2) getting skills to reach the different possible symbolic state changes is desirable. These are not practical in most robotic domains, where even the symbolic state space might be extremely large and only a small subset of symbolic state changes might be important for useful skills. Thus, it seems difficult if not impossible to apply this work to continuous, long-horizon problems beyond the relatively simple setting (physically-embedded games with small and simple symbolic state spaces) that the authors have chosen.
- The task performance evaluation is limited to tasks that only require up to 5 moves. This seems like an unjustified choice that simplifies the task and significantly reduces the horizon. Given that the approach was trained to generalize to settings regardless of horizon, why was the horizon limited?

**Summary Of Recommendation:**

Overall, I think this is an interesting, well-written paper that provides an interesting new direction for hierarchical RL. However, the limited applicability and setting make it seem rather narrow in scope compared to existing hierarchical RL approaches. I think it is valuable and interesting to the community, but could be made significantly better if the authors clearly address and point out the applicability of their method in the writing of the paper or via additional environments/domains in the experimental evaluation.

---

> ### Author Response · Authors · 2022-08-26
> **Author response**
>
> Thank you for your valuable suggestions on our work!
> In the following, we discuss your questions. We also updated our paper with your suggestions.
>
> ### Assumption: Abstract symbolic state space is small
> Please note that the cardinality of the state space, i.e. number of all symbolic states, is large. Our binary state representations admits 2^25 and 2^81 board configurations for LightsOut and TileSwap. However, not all board configurations are feasible or can be reached at a specific solution depth.
> Please refer to Table 2 in the supplementary material for all possible board configurations for various solution depths.
>
>
> ### Assumption: Getting skills to reach the different possible symbolic state changes is desirable
> It is indeed an open research question how our skill exploration objective could be combined with demonstrations or task-specific objectives to guide the search for symbolic actions and limit the search space. We will discuss this in the limitations section.
>
> ### Limited horizon: BFS speed
> The branching factors of the LightsOut and TileSwap planning problems are the number of skills 12 and 25.
> Runtime complexity of BFS is linear in the size of the search graph which makes planning time too large for solution depths larger than 5 in our experiments.
> We refer to sec. S.5 in the updated supplementary material for an analysis of planning times for different solution depths (up to depth 9) on the LightsOutCursor environment, showing that our agent can generalize to longer solution depths.
> Future work could investigate more sophisticated heuristics and pruning strategies.
>
> ### Further comments:
> We revised the paper to explicitly state that the number of skills K is being provided as input.
> Please observe the "more skills" ablation in Fig. 3 which demonstrates that our method can also work with an overestimate of the number of skills.

---

### Official Review · Reviewer_rgwq · 2022-08-03

**Originality:** Very Good
**Technical Quality:** Excellent
**Clarity Of Presentation:** Excellent
**Impact:** 4

**Recommendation:**

Strong Accept: I recommend accepting the paper and will argue for my recommendation even if other reviewers hold a different opinion.

**Summary:**

This paper presents a method that jointly learns a predictive model in the symbolic level and a skill-conditioned policy.  After training, the learned predictive model allows the authors to do long-horizon planning in the high level, and execute those plans using the learned policy. The method is both evaluated in simulation and on a real robot.

**Issues:**

No issues.

**Quality Of The Limitations Section:**

Limitations are addressed clearly

**Reviewer Expertise:**

5: The reviewer is absolutely certain that the evaluation is correct and very familiar with the relevant literature

**Robotics Focus:**

Sufficient demonstration on hardware

**Strengths And Weaknesses:**

## Strengths
Automatically learning a discrete set of skills as well as a transition function in high level is extremely interesting. The paper shows interesting results that in many cases, the method can recover all the discrete skills needed to solve the task.

## Weaknesses and questions
-  about the assumption

The authors have made it clear that there is one assumption of this work, which is that the symbol definition and the mapping from state to symbol are given. While I consider this assumption reasonable, it dramatically simplified the problem, thus posing an issue of applying this method to other tasks.

-  about the efficiency

From the experiments, we can see as the task becomes more complex, recovering all skills becomes harder. It seems to me that the method in this paper is still not efficient enough to be scaled up to more interesting robotic tasks (with a magnitude of more symbolic states and skills). I'm curious about the authors' view on learning efficiency. What bottlenecks, which prevent the system to discover skills more efficiently, does this algorithm have? One reason, as mentioned in the relabelling section, is that the coupled nature of the forward model and the policy make the learning in early stage hard. Another reason I can think of is the exploration issue. Does the method have trouble in encouraging exploration?

-  about the learned skills

The training objective encourages the skill to reach diverse successor states, and be consistent with the forward model. Are there any constraints on the skills being consistent over different starting states?  For example, in two different starting states where different lights are on, if one skill is learned in the first starting state to be pressing the up-left button, will this skill also be pressing the same button if the initial lights and robot state are different? Figure S1 suggests that the skills are consistent. But I wonder how this is achieved.

**Summary Of Recommendation:**

This paper bridges the gap between the neural network based reinforcement learning and symbolic planning. This could potentially have a large impact on the direction of learning more robust and interpretable models.

---

> ### Author Response · Authors · 2022-08-26
> **Author response: Part 2**
>
> ### Learned skills
>
> In principle, our method could assign different behaviors to each skill depending on the symbolic state $z_0$, as both the low-level policy and the forward model are conditioned on the initial symbolic state.
> In both environments, skills are typically learned which push a particular field (LightsOut) or swap tiles between two particular fields (TileSwap) irrespective of the symbolic board state (toggle state of Lights or distribution of tiles).
> SEADS seems to favor generalizing skill policies between symbolic states if knowledge of the symbolic states is not necessary to execute the skills on the low level.
>
> In the following, we provide arguments and experiments, that the SAC agent and the forward model still make use of the symbolic state if required.
>
> (i) For LightsOut, the forward model would still be able to make accurate predictions if the predicted bit flip probability only depends on the skill index $k$, i.e., $p_{flip} = f_\theta(k)$.
> However, to model the symbolic transitions of TileSwap accurately, this simplification does not work.
> The symbolic state is needed to determine which tiles (varying bit locations) need to be swapped at the pushed location.
> As we observe that our agent can solve the TileSwap environments with high success rate, we conclude that our model can cover the general case that the bit flip probabilities depend on $z_0$.
>
> (ii) We demonstrate that the skill policies $\pi$ can also learn dependencies on the symbolic state $z_0$.
> To this end, we performed experiments on a slightly modified LightsOut environment. In this environment, the state of a field and its neighbors can only be toggled if (a) for a switched-on field, the push happens in the lower half of the field, (b) for a switched-off field, the push happens in the upper half. In this setting, to reliably modify the symbolic state through toggling a field, a skill has to take the current illumination state of the field into account (which is part of the symbolic state). We trained our SEADS agent on this environment for 500k environment interactions and attached a plot visualizing the different skills for (a) all fields switched off and (b) all fields switched on. Please find a visualization of the skill trajectories attached. We can clearly see that for the switched-off fields, the skills push the upper halfs of the fields, while for the switched-on fields, the skills push the lower halfs. By this, we dismiss the hypothesis that our SAC agent is unable to learn behavior which depends on the symbolic state $z_0$.
>
> In conclusion, the observed skill consistency seems to be a consequence of inductive biases in our forward model and SAC agent, presumably being that it is the simplest option for the SAC agent to only take the skill index $k$ into account and discard the initial state $z_0$, if possible.

---

> ### Author Response · Authors · 2022-08-26
> **Author response: Part 1**
>
> We thank the reviewer for the detailed feedback on our paper.
> We discuss your questions in the following. Your suggestions have been incorporated in the updated paper.
>
>
> ### Assumption: Symbolic representation available
>
> We agree that learning the full symbolic representation including the state abstraction is a very interesting problem.
> However, we demonstrate that even when *knowing* the symbolic state representation, jointly learning a symbolic forward model and accompanying low-level skills is not trivial and requires careful design of the learning algorithm.
> In addition, as mentioned in the paper and as reviewer w2Y9 points out, symbolic learning methods are available to learn a symbolic representation separately.
> It is very interesting future work to extend the presented method by a component which allows for learning the symbolic representation.
>
>
> ### Efficiency
>
> Exploration happens through the injected noise on actions in SAC. New skills are found by random perturbations of actions and observing new symbolic state transitions. This works well in board games, since the board fields affected by the skills are spatially close. A skill of pushing a field does not need to be learned from scratch, but can be learned by perturbation of a skill that pushes a neighboring field. More complex environments could require additional techniques to explore the unexplored regions of the low-level state space more efficiently. In the updated supplementary material (sec. S4.2) we report experimental results for variants of the LightsOutCursor environment which are more challenging in terms of exploration. We increased the board size and introduced a spacing between fields, making detection of new skills from random execution of already learned skills less likely. While for board sizes up to 9x9 a large majority of skills is detected (75.3/81 for 9x9), on the 13x13 environment only 118.3/181 skills were detected after training for 1.5M environment interactions.

---

### Official Review · Reviewer_w2Y9 · 2022-08-05

**Originality:** Very Good
**Technical Quality:** Very Good
**Clarity Of Presentation:** Very Good
**Impact:** 4

**Recommendation:**

Strong Accept: I recommend accepting the paper and will argue for my recommendation even if other reviewers hold a different opinion.

**Summary:**

This work proposes a skill discovery method given a set of symbolic states (state abstractions). The method consists of two parts. In the first part, a low-level controller, a = pi(s_0, k), which can be regarded as an option, outputs continuous actions to move the robot to the terminal state, s_T, given the initial state s_0, and the option index k. Here, s_0 and s_T correspond to symbolic states z_0 and z_T, respectively. In the second part, there is a symbolic forward model, z_T = q(z_0, k), which outputs the next symbolic state z_T given the current symbol z_0 and the option index k. By maximizing the mutual information between z_T and k, the robot learns a set of options that are both diverse and predictable given the initial symbol z_0. The low-level controller is trained by an RL algorithm (SAC), and the symbolic forward model is trained in a supervised manner.

**Issues:**

Although the related work section is quite extensive and good read, the  authors missed a very relevant literature. A number of recent work (Silver et al., 2021&2022; Chitnis et al., 2021) also works on skill learning where skills are represented with both symbolic and low-level components and hierarchical learning is performed. These studies use demonstrations rather than Reinforcement Learning, but otherwise have very similar assumptions, approaches and goals. The authors need to include one or more of these studies in their paper.

Silver, Tom, et al. "Learning Neuro-Symbolic Skills for Bilevel Planning." arXiv preprint arXiv:2206.10680 (2022).
Chitnis, Rohan, et al. "Learning neuro-symbolic relational transition models for bilevel planning." arXiv preprint arXiv:2105.14074 (2021).

- line 132: please define k before its use.
- line 152: d and are is mixed up.
- According to line 184, a change in state might not be observed in an episode. Then, how sound is it to include such an episode for FM learning (line 192)?
- BFS does not give optimal solution if the actions costs are not same. Because the skills are discovered by the robot, each skill might have different length or cost. Therefore, the solutions might not be optimal. Please comment.
- line 260: In 2D simulated environment, is the robot really torque controlled?
- Although the notation is consistent and sound in Section 3, the use of numbers/labels/brackets in the subsecripts is a bit confusing. Is it possible to improve?
- In task performance evaluation in Section 4, do environment steps refer to the number of optimization steps or the number of data points (i.e., environment interactions)? This is not clear because 10m steps are needed for experiments in the simulation while it is only 5000 interactions in the real world.
- It would be nice to show what kind of skills are discovered after training (i.e., a scatter-plot of executions from an initial point for option k), and the average number of low-level actions for each skill. Regarding the statement, "The board coordinates are x, y ∈ [0, 1], the maximum displacement per timestep is ∆x, ∆y = 0.2.", if each displacements after each action become very high, then it would be enough to output only 3 or 4 low-level actions.
- Moreover, for the reward improvement, I think having multiple high-level actions that moves to the same terminal state from the same initial state (i.e., not unique) might be a feature in other settings in which the robot learns two modes of operations and selects one at different conditions.


**Quality Of The Limitations Section:**

Limitations are addressed clearly

**Reviewer Expertise:**

4: The reviewer is confident but not absolutely certain that the evaluation is correct

**Robotics Focus:**

Sufficient demonstration on hardware

**Strengths And Weaknesses:**

The proposed method is a novel contribution for discovering skills given state abstractions. Although the assumption of state abstractions is strong, these abstractions can be indeed learned via symbol learning methods also mentioned in the text [24, 25, 27, 28, 29]. The idea of concatenating an option index k to the state, and maximizing the mutual information between k and z_T is very interesting. The paper is nicely written, explaining each definition thoroughly. Experiments are well-designed and there is an ablation study for each part of the method.

One slight downside might be that the method is only tested in 2d environments; it would have been good to see the performance of the method in a scaled environment, say, a 3d version of the lights out, since the exploration of the environment might not be as easy as in 2d. Of course, this is more related to exploration, but it would still be interesting to see the limits of the method.

**Summary Of Recommendation:**

I believe the contribution of the paper is very novel and significant. Experiments show the applicability of the method both in simulated and real-world environments. The paper is well-written, and the accompanying video also helps assessing experiments.

---

> ### Author Response · Authors · 2022-08-26
> **Author response: Part 1**
>
>
> Thank you for your effort and time to compose a very detailed and helpful review!
> We updated our paper with your suggestions.
>
> ### Cf. "3D Lightsout"
>
> We tested our approach on a 3D version of LightsOut, and added the results to the supplementary material (please refer to the updated pdf in the attachment for the results, sec. S.4.1). In this variant the fields are elevated/recessed depending on their distance to the board's center. This poses an additional challenge to the agent, as it has to avoid to push fields with its fingers accidentally during skill execution. The increased complexity over LightsOutJaco leads to a slightly lower average number of found skills. The planning success rate reduces to 80%, compared to 97.6% for LightsOutJaco (evaluation protocol as in main paper).
> In terms of exploration, our approach is currently limited by the exploration strategy implemented by the SAC agent which perturbs actions randomly using a Gaussian distribution.
> New skills are found by random perturbations of actions and observing new symbolic state transitions. This works well in board games, since the board fields affected by the skills are spatially close. A skill of pushing a field does not need to be learned from scratch, but can be learned by perturbation of a skill that pushes a neighboring field. More complex environments could require additional techniques to explore the unexplored regions of the low-level state space more efficiently.  In the updated supplementary material (sec. S4.2) we report experimental results for variants of the LightsOutCursor environment which are more challenging in terms of exploration. We increased the board size and introduced a spacing between fields, making detection of new skills from random execution of already learned skills less likely. While for board sizes up to 9x9 a large majority of skills is detected (75.3/81 for 9x9), on the 13x13 environment only 118.3/181 skills were detected after training for 1.5M environment interactions.
>
>
> ### Cf. "Related work"
> Thank you for the interesting pointers to related work which we have incorporated into the paper.
> As mentioned these works do not use intrinsic reward to learn the symbolic action representations by exploration, but learn from demonstrations.
>
>
> ### Cf. "Issues"
> > l. 132/ l. 152: Thank you, we have updated the paper accordingly.
>
> > Excluding no-op trajectories in forward model training
>
> Including no-op trajectories in the forward model training allows for accurately reflecting the policies' behavior in the forward model. The policy should only receive a high intrinsic reward only if it *consistently* changes the symbolic state; if it occasionally fails to change the symbolic state, the intrinsic reward should be lower. As the intrinsic reward is computed using the forward model (eq. 1), the forward model also needs to be trained on "faulty" executions in which the symbolic state does not change.
>
> > BFS optimality
>
> You are right, our current BFS planner is optimal with respect to the "number of skills" executed within a plan, but does not take into account that each skill may have different costs (e.g., number of environment interactions or other control costs). We have added discussion in "Assumptions and Limitations".
>
> > Reacher torque control
>
> We use the "Reacher" manipulator from the `dm_control` suite, which uses Mujoco's `<motor>` actuators to control the joints, which are torque controlled.
>
> > Notation
>
> We experimented with other notations and converged to the current one as our perceived "local optimum", but are very open to suggestions about different notations.
>
> > Environment steps
>
> Yes, throughout the paper, "environment steps" consistently refers to "environment interactions". The real-world experiment needs a comparably small number of interactions as the SEADS agent sends push locations to the robotic arm instead of low-level joint torques/velocities/positions as in the "Reacher" and "Jaco" experiments. To make the experiment more closely match the simulated environments, we have added an additional experiment reported concisely in the main paper and detailed in the supplementary material (see sec. S.5.2), in which we do not send push locations to the robot, but rather displacements in the x-y-plane, as in the "Cursor" environments. Here, we observe that the agent is able to solve the posed LightsOutGame with high success rate after $\approx 160.000$ environment interactions, which is in a similar range as the results reported on the Cursor environments (see Fig. 3).
> We will update the main paper and the accompanying video accordingly.

---

> > ### Author Response · Authors · 2022-08-26
> > **Author response: Part 2**
> >
> >
> > > Skill trajectories / Number of low-level actions
> >
> > Thank you for the suggestion to provide insights into the properties of the learned skills, e.g., how many steps they take and what trajectories they follow. We report the distribution of skill length in section S2.3 of the updated supplementary. Your estimate about the skill length for the Cursor environments is quite accurate, the median skill length is 3 for LightsOutCursor and 2 for TileSwapCursor. For the Reacher and Jaco environments the median skill length is higher with $\approx$ 10 steps per skill. The trajectories followed by the learned skills are depicted in Figures S.2-S.4.
> >
> > > Multiple high-level actions
> >
> > It is a design choice in our approach to learn diverse symbolic actions which would require uniqueness if the number of skills matches the possible skills in the environment. Handling ambiguous settings for skill execution in our approach could be achieved by learning stochastic skill policies which represent multi-modality and depend on the low-level environment state (for instance moving left or right around an obstacle to reach to a location).

---

### Meta-Review · Area_Chair_S1uN · 2022-08-14

**Recommendation:** Accept (Oral)
**Confidence:** 4

**Metareview:**

Reviewers like the novelty of the formulation and the good writing. Shared concerns include the limited experiments and the contrived setup. Other concerns include those on generalization and related work.  Please consider addressing these points.

The authors did a good job during the rebuttal period, after which all reviewers were in favor of acceptance.  The AC agrees.  Please revise the paper accordingly based on the reviews.

**Best Paper Nomination:**

No